# Study of Effects of MoS₂ Nanofluid MQL Parameters on Cutting Forces and Surface Roughness in Hard Turning Using CBN Insert

**Ngo Minh Tuan [1] , Tran The Long [1] and Tran Bao Ngoc [2],***

1 Department of Manufacturing Engineering, Faculty of Mechanical Engineering,
Thai Nguyen University of Technology, Thai Nguyen 250000, Vietnam; minhtuanngo@tnut.edu.vn (N.M.T.);
tranthelong@tnut.edu.vn (T.T.L.)
2 Department of Fluids Mechanic, Faculty of Automotive and Power Machinery Engineering,
Thai Nguyen University of Technology, Thai Nguyen 250000, Vietnam
* Correspondence: baongoctran@tnut.edu.vn; Tel.: +84-397-375-897

**Abstract:** Lubrication and cooling in hard machining is an urgent and growing concern. The use of a suitable cooling lubrication condition is a crucial factor and has a great influence on the machining efficiency and the machined surface quality in hard machining. Among the proposed technological solutions, minimum quantity lubrication (MQL) using nano-cutting oils is a novel solution and its effectiveness has been proven for hard turning. This work aims to investigate the influence of MQL technological parameters using MoS₂ nano-cutting oil including nanoparticle concentration, air pressure, and air flow rate on surface roughness and the resultant cutting force in hard turning using CBN inserts. Box-Behnken optimal experimental design and ANOVA analysis were used to study the influence of the input parameters and determine the optimal values. The results present the influence of the survey parameters and provide technological guides for specific objective functions for further sustainable studies on MQL hard turning using nano-cutting oil.

**Keywords:** hard machining; hard turning; MQL; MoS₂ nanoparticles; nanofluid; nano-cutting oil; difficult-to-cut material; air pressure; air flow rate





## 1. Introduction

Along with the rapid development of materials technology and especially nanomaterials, it is now not difficult to see the applications of nanomaterials in our lives and in industrial fields [1]. There is no denying that nanomaterials have been drastically changing different industrial fields. In recent years, with the increasing demand for lubrication and cooling in the cutting zone in machining metals, especially for hard and difficult-to-cut materials [2], there are many proposed technological solutions to improve the efficiency of the cutting process. Furthermore, the rapid and powerful development of hard machining technology has put more and more pressure on lubrication and cooling technologies. High cutting forces and heat are still the major challenges in machining hard materials, and they are closely related to flank wear. In hard machining, tool flank wear is a parameter that is very sensitive to the cutting force and the quality of the machined surface, which determines the tool life [3]. Therefore, cooling lubrication in the cutting zone plays a crucial role [4].

The conventional solution is the flood coolant method, which has disadvantages such as limited lubrication and environmental pollution, so the use of this technology in sustainable and environmentally friendly production will be limited in the future [5]. Therefore, lubricating and cooling technologies have been researched and developed, including technologies such as minimum quantity lubrication (MQL) and minimum quantity cooling lubrication (MQCL) using nano-cutting oils. The MQL method shows high lubricating efficiency, with the cutting oil directly sprayed into the cutting zone in the form of a mist [6]. In

addition, with a very small amount of cutting oil used, it is classified as an environmentally friendly technology. However, the use of MQL has been shown to be effective when machining steels before heat treatment or materials with low hardness [7]. When machining hard materials, MQL shows poor performance due to a limited cooling capacity [8]. To overcome this drawback, MQCL technology was developed to improve the cooling efficiency, but mainly studies are using cutting oils with good cooling properties such as emulsion oil (to produce the cooling effect) for MQL technology [9–11]. Recently, there have been some studies on using the MQCL method for machining hard materials. Dong et al. [12] used the MQCL nozzle based on the principle of the Ranque–Hilsch vortex tube for hard milling of steel SKD 11 (60–62 HRC). The cutting oil comes out of the MQCL nozzle in the mist form at high pressure and low temperature (4–8 °C), creating high lubricating and cooling efficiency. The obtained results show that hard milling performance is significantly improved and the machinability of carbide inserts is enhanced. In addition to that, a novel solution to improve lubrication and cooling efficiency is to use nano-cutting oil for MQL technology. Moreover, the trend of using vegetable oils and recycled oils as MQL-based cutting fluids to overcome the environmental issues has gained much attention from researchers around the world. O. Pereira et al. [6] deeply studied the performance of four different sunflower oils (with and without a biodegradable antioxidant additive), castor oil, canola oil, and ECO-350 recycled oil in MQL end milling of the Inconel 718 alloy. Among the investigated oils, ECO-350 recycled oil and canola oil present the lowest viscosity, which contributes to achieving better oil penetration into the cutting zone. An extension of the tool's life by about 15% and 30% was reported by using high oleic sunflower oil and ECO-350 recycled oil, respectively. Rahman et al. [13] studied the turning process of the Ti-6Al-4V alloy within an MQL environment using $Al_2O_3$, $MoS_2$, and $TiO_2$ vegetable-based nano-cutting oils. The results showed that the efficiency of the cutting process is improved due to the enhanced lubricating and cooling capacity by using nano-cutting oils. An analysis of the machined surface microstructure indicated that the surface quality is better when using 0.5% $Al_2O_3$ nano-cutting oil based on rapeseed oil. Hegab et al. [14] investigated the MQL turning of Inconel 718 using multi-walled carbon nanotubes (MWCNTs) and $Al_2O_3$ nanoparticles suspended in rapeseed oil. The author concluded that the tribology of MWCNTs nano-cutting oil is better than that of $Al_2O_3$. In addition, the presence of nanoparticles contributed to improving the lubrication and cooling efficiency in the cutting zone. Darshan et al. [15] studied the effects of $Al_2O_3$, $MoS_2$, and graphite sunflower-based nano-cutting oils on turning of the Inconel 800 alloy. From the experimental results, the authors pointed out that graphite and $MoS_2$ nano-cutting oils had better efficiency compared to $Al_2O_3$ nano-cutting oil due to better thermal conductivity and lubrication performance. In a similar study, Gupta et al. [16] recorded the machining performance of the turning process of the Inconel 800 alloy, a difficult-to-cut material, significantly improved with the use of nano-cutting oils in the MQL technique, and a very small amount of vegetable oil was used, which contributed to the reduction in the maximum negative impact on the environment due to its biodegradable characteristic.

Some methodologies are commonly used to study the influence, predict, and optimize the parameters of the cutting mode such as regression analysis (RA), Box-Behnken, response surface methodology (RSM), artificial neural network (ANNs), and genetic algorithms (GAs). F.J. Pontes et al. [17] used radial base function (RBF) neural networks combined with Taguchi's orthogonal array to predict the surface roughness, $R_a$, in hard turning of SAE 52100. The authors concluded that RBF neural networks with the application of design of experiment (DOE) methodology were suitable and effective for $R_a$ prediction in the hard turning process. ANN design should be simplified in order to be widely used in further studies. G. Kant and his co-authors [18] built models based on artificial neural networks and genetic algorithms to predict and optimize the cutting parameters for the hard turning process for minimum surface roughness. The obtained results indicated that the predicted surface roughness values were in good agreement with the validation machining tests. Furthermore, the authors found that the ANN results achieved a better performance compared to regression and fuzzy logic

models. U.M. R. Paturi et al. [19] applied RA and ANN methodologies to predict the surface roughness of hard turning of AISI 52100. The Taguchi L27 orthogonal array was used to build the experimental planning design for the input variables, including cutting speed, feed, and depth of cut. A good agreement between the predicted and experimental values was reported, but the ANN model showed better results. J. P. Maran et al. [20] compared artificial neural network and response surface methodology (RSM) modeling in studying mass transfer prediction. The authors concluded that ANN models can bring out more accurate predictions than RSM models, but the RSM with Box-Behnken design is very useful to present the individual/interaction influences and contribution of the investigated factors from the coefficients in the regression models. The study of cutting parameters, MQL technological parameters, and especially MQL using nano-cutting oil is an up-to-date topic, so the study on the influence of each input variable and their interaction effects on the objective function is very important because the presence of different types of nanoparticles in the cutting oil will create different lubrication and cooling mechanisms.

Each type of nanoparticle has different properties and shapes, so the nanoparticle type and concentration in the base cutting oil are very important parameters for the lubricating and cooling efficiency. $Al_2O_3$ nanoparticles are reported to have high hardness, strength, and near-spherical morphology [21]. Therefore, when they penetrate into the cutting zone, it will contribute to convert the sliding friction into rolling friction, thereby reducing the coefficient of friction and cutting forces. $MoS_2$ and graphite nanoparticles have a layered structure and thermal conductivity coefficient higher than that of $Al_2O_3$ nanoparticles [22], and they have good lubricating properties. Therefore, these two types of particles are widely used to form nano-cutting oils in metal cutting. Zhang et al. [22] investigated the efficiency of different vegetable oils with $MoS_2$ nanoparticles as the base cutting oil for MQL grinding. A reduction in the heat and grinding force was reported, which proves the better lubricating effect, thereby expanding the application of vegetable oils in machining as well as minimizing the adverse effects on the environment. Uysal et al. [23] made a study on the milling process under the MQL environment using vegetable oil with $MoS_2$ nanoparticles. The obtained results show that the machined surface quality and tool life were improved due to the excellent lubricating and cooling effects of $MoS_2$ nano-cutting oil. Ayşegül Yucel et al. [24] conducted an investigation on the effects of $MoS_2$ nano-cutting oil based on mineral oil for turning AA 2024 T3 aluminum alloy. The authors found an enhancement in surface roughness and surface topography as well as a significant reduction in the built-up-edge (BUE) formation when compared with dry turning. Furthermore, the cutting temperature and tool wear were reduced in comparison with dry and pure MQL conditions.

Moreover, an outstanding property of $MoS_2$ nanoparticles is that they have a large surface area, which tends to form tribofilm when using a reasonable nanoparticle concentration, air pressure, and air flow rate in the MQL/MQCL environment [25,26]. However, through a literature review, it can be clearly seen that the number of studies investigating and determining these technological parameters is still very limited, especially for the hard turning process [27]. Therefore, the authors are motivated to study the influence of $MoS_2$ nanoparticle concentration, air pressure, and air flow rate in hard turning of 90CrSi steel (60–62 HRC) under the MQL condition using CBN inserts.

## 2. Materials and Methods

A diagram of the experimental setup for this research is shown in Figure 1. The MQL nozzle is arranged to spray directly onto the flank face of the cutting tool [28]. Compressed air is supplied by the air compressor and regulated by the pressure regulator valve and the air flow rate is regulated by the flow control valve. Nano-cutting oil is put into the MQL nozzle according to the capillary principle. When meeting a high-pressure air flow, the cutting oil coming out of the nozzle will be in the mist form. The dynamometer is set up under the tool holder and is connected to an amplifier and signal converter and a computer installed with the Dasylab version 10.0 data acquisition software released in 2007 (Measurement Computing Corporation, 10 Commerce Way Norton, MA, USA).

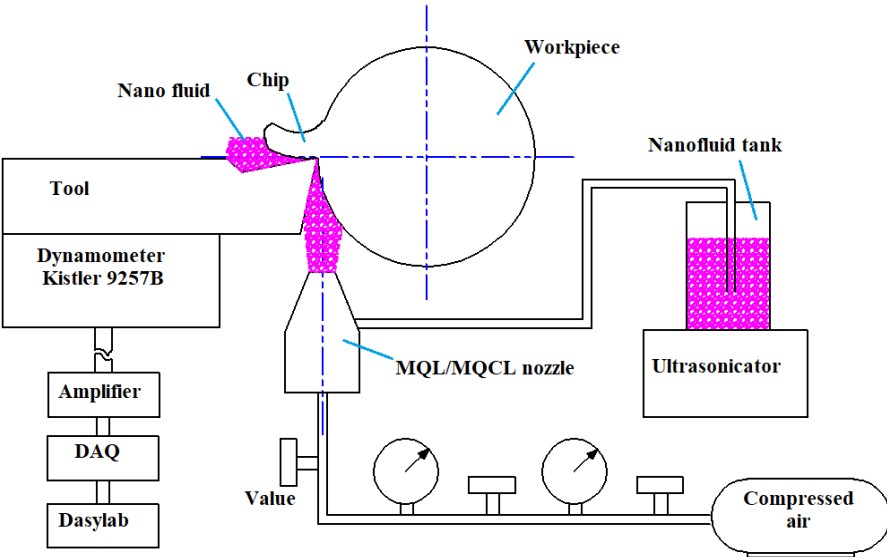

**Figure 1.** Diagram of experimental setup.

The samples were 90CrSi hardened steel (60–62 HRC) with the chemical composition shown in Table 1. Sandvik CBN inserts with the designation of CCGW09T308S01020FWH7025 were used. The $MoS_2$ nanoparticles were suspended in rice bran oil to form nano-cutting oils with three different concentrations (0.2%, 0.5%, and 0.8%); the preparation of nano-cutting oils was discussed in [29]. The cutting parameters were fixed at: cutting speed, V = 160 m/min; depth of cut, t = 0.12 mm; and feed rate, f = 0.12 mm/rev. The cutting force values were directly measured during the machining process. The surface roughness was measured three times using SJ210 of Mitutoyo, Japan after each cutting trial and the average value was taken.

**Table 1.** Chemical composition in % of 90CrSi steel (Republished from [29]).

| Element | C | Si | Mn | Ni | S | P | Cr | Mo | W | V | Ti | Cu |
|---|---|---|---|---|---|---|---|---|---|---|---|---|
| Weight (%) | 0.85–0.95 | 1.20–1.60 | 0.30–0.60 | Max 0.40 | Max 0.03 | Max 0.03 | 0.95–1.25 | Max 0.20 | Max 0.20 | Max 0.15 | Max 0.03 | Max 0.3 |

Box-Behnken optimal experimental design was used to build an experimental planning diagram with three input parameters (Table 2) in order to study the effects of nanoparticle concentration (NC), air pressure (*p*), and air flow rate (*Q*) on the surface roughness ($R_z$) and resultant cutting force ($F_r$), which is determined by Equation (1) below:

$$F_r = \sqrt{F_x^2 + F_y^2 + F_z^2} \tag{1}$$

**Table 2.** Input parameters and their values.

| No. | Parameter | Symbol | Low Level | High Level | Responses |
|---|---|---|---|---|---|
| 1 | Nanoparticle concentration (%) | NC | 0.2 | 0.8 | |
| 2 | Air pressure (bar) | *p* | 4 | 6 | Surface roughness, $R_z$ Resultant cutting force, $F_r$ |
| 3 | Air flow rate (L/min) | Q | 150 | 250 | |

Currently, there have been many methods used to analyze and optimize machining processes. Many studies have used optimization algorithms such as ANN, GA, and so on, to provide a prediction for the regression function and determine the optimal set of parameters. However, these methods do not evaluate the influence of the survey variables and their interaction effects on the objective function. In addition, optimization methods using experimental planning models such as Taguchi, Box-Behnken, factory, central composite design (CCD), etc., are commonly used in experimental research because they can reduce

the number of experiments and still evaluate the influence of the factors on the objective function. At the same time, these methods can also be used to find predictive models and determine the optimal set of parameters for the objective function. Furthermore, the Box-Behnken experiment design is usually used only when the number of factors to be investigated is greater than 2. The Box-Behnken experiment design for k factors is a combination of the full two-level experimental design for p factors ($p < $k) and the block design is not complete. Thus, in each treatment there will be p factors with values of high level (+1) or low level (−1), the remaining factors are at the center level (0), as shown in Figure 2.

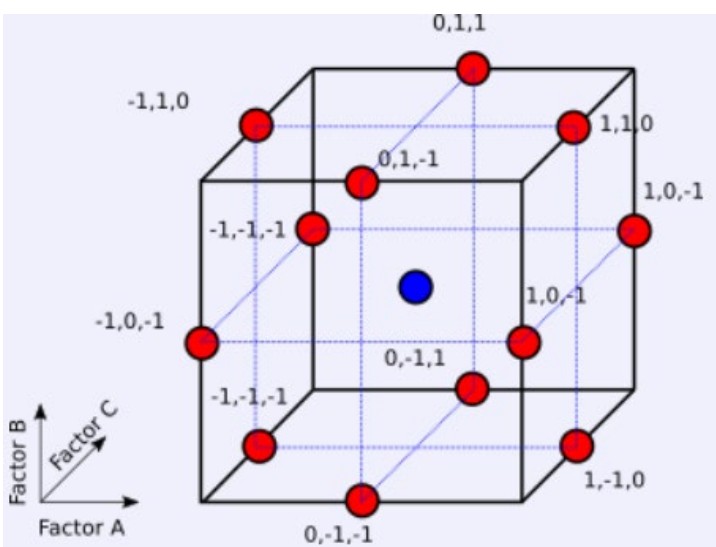

**Figure 2.** Box-Behnken experiment design.

Accordingly, the Box-Behnken optimal experimental design was selected and applied. The Minitab 19 software was used to build the matrix of 15 experiments according to Box-Behnken optimal experimental design for the 3 investigated variables. The experiment trials were carried out by following the RunOrder and the measured values of $R_z$ and $F_r$ are shown in Table 3.

**Table 3.** Experimental design and measured responses (cutting speed, V = 160 m/min; depth of cut, t = 0.12 mm; and feed rate, f = 0.12 mm/rev).

| Std. Order | Run Order | PtType | Blocks | NC (wt%) | $p$ (bar) | Q (L/min) | $R_z$ (μm) | $F_r$ (N) |
|---|---|---|---|---|---|---|---|---|
| 5 | 1 | 2 | 1 | 0.2 | 5 | 150 | 1.848 | 307.1019 |
| 3 | 2 | 2 | 1 | 0.2 | 6 | 200 | 1.980 | 385.1642 |
| 14 | 3 | 0 | 1 | 0.5 | 5 | 200 | 1.198 | 135.7268 |
| 10 | 4 | 2 | 1 | 0.5 | 6 | 150 | 1.321 | 165.9382 |
| 13 | 5 | 0 | 1 | 0.5 | 5 | 200 | 1.203 | 143.3989 |
| 12 | 6 | 2 | 1 | 0.5 | 6 | 250 | 1.353 | 190.324 |
| 7 | 7 | 2 | 1 | 0.2 | 5 | 250 | 1.955 | 302.4206 |
| 8 | 8 | 2 | 1 | 0.8 | 5 | 250 | 1.898 | 658.5913 |
| 11 | 9 | 2 | 1 | 0.5 | 4 | 250 | 1.291 | 208.8865 |
| 6 | 10 | 2 | 1 | 0.8 | 5 | 150 | 2.518 | 728.2449 |
| 4 | 11 | 2 | 1 | 0.8 | 6 | 200 | 2.143 | 489.8893 |
| 1 | 12 | 2 | 1 | 0.2 | 4 | 200 | 1.295 | 268.6225 |
| 2 | 13 | 2 | 1 | 0.8 | 4 | 200 | 2.436 | 537.0321 |
| 9 | 14 | 2 | 1 | 0.5 | 4 | 150 | 1.846 | 141.9083 |
| 15 | 15 | 0 | 1 | 0.5 | 5 | 200 | 1.213 | 155.6953 |

## 3. Results and Discussion

### 3.1. Effects of NF MQL Parameters on Surface Roughness

ANOVA analysis was performed by using the Minitab 19 software (Minitab Inc., State College, PA, USA) with the significance level $\alpha = 0.05$. The results of the ANOVA analysis are shown in Table 4. The analysis results show that the Fisher coefficient value for the model is quite large, at 10.34, and the *p*-value is very small when compared to 0.05, which proves that the predictive model is consistent and less affected by noise. From this, it can be seen that among the investigated factors, nanoparticle concentration (NC) and the interactions NC*NC and NC*p have the strongest influences on the objective function $R_z$. The degree of influence of the survey variables and their interactions on the surface roughness value, $R_z$, is also shown on the Pareto chart (Figure 3). Factors with influence coefficients larger than the reference influence coefficient (2.571) strongly affect the objective function. The regression model for $R_z$ is given below:

$$Rz~(\mu m) = 7.62 - 0.26*NC - 0.973*P - 0.0384*Q + 7.56*NC*NC + 0.0784*P*P + 0.000068*Q*Q - 0.815*NC*P - 0.01212*NC*Q + 0.00294*P*Q \quad (2)$$

The fit of the regression model is evaluated through the coefficient of determination $R^2$. The results show that the coefficient of determination $R^2 = 94.90\%$ and the adjusted coefficient of determination $R^2 = 85.72\%$ is quite large, which proves that the regression model is suitable with the experimental data.

**Table 4.** Result of ANOVA analysis for surface roughness, $R_z$.

| Source | DF | Adj. SS | Adj. MS | F-Value | *p*-Value |
|---|---|---|---|---|---|
| Model | 9 | 281.326 | 0.31258 | 10.34 | 0.010 |
| Linear | 3 | 0.59415 | 0.19805 | 6.55 | 0.035 |
| NC (wt%) | 1 | 0.45936 | 0.45936 | 15.19 | 0.011 |
| P (bar) | 1 | 0.00063 | 0.00063 | 0.02 | 0.891 |
| Q (L/min) | 1 | 0.13416 | 0.13416 | 4.44 | 0.089 |
| Square | 3 | 176.171 | 0.58724 | 19.42 | 0.003 |
| NC (wt%)*NC (wt%) | 1 | 170.942 | 170.942 | 56.53 | 0.001 |
| P (bar)*P (bar) | 1 | 0.02270 | 0.02270 | 0.75 | 0.426 |
| Q (L/min)*Q (L/min) | 1 | 0.10629 | 0.10629 | 3.52 | 0.120 |
| 2-Way Interaction | 3 | 0.45740 | 0.15247 | 5.04 | 0.057 |
| NC (wt%)*P (bar) | 1 | 0.23912 | 0.23912 | 7.91 | 0.037 |
| NC (wt%)*Q (L/min) | 1 | 0.13213 | 0.13213 | 4.37 | 0.091 |
| P (bar)*Q (L/min) | 1 | 0.08614 | 0.08614 | 2.85 | 0.152 |
| Error | 5 | 0.15119 | 0.03024 | | |
| Lack-of-Fit | 3 | 0.15108 | 0.05036 | 863.29 | 0.001 |
| Pure Error | 2 | 0.00012 | 0.00006 | | |
| Total | 14 | 296.446 | | | |

The effect of each studied factor on the average values of surface roughness was analyzed and is shown in Figure 4. The results show that the concentration of $MoS_2$ nanoparticles strongly influenced the surface roughness values. When increasing the nanoparticle concentration from 0.2% to nearly 0.45%, the surface roughness value gradually decreases. However, if the $MoS_2$ nanoparticle concentration continues to increase to 0.8%, the $R_z$ values increase sharply. The main reason is that in the range of NC = 0.2–0.45%, $MoS_2$ nanoparticles penetrate into the cutting zone and form tribofilm, which helps to reduce friction, thereby decreasing surface roughness. When the concentration of $MoS_2$ nanoparticles increases to 0.8%, which is inappropriate [25], it leads to the phenomenon of

compression among nanoparticles and adhesion on the cutting edge [12,25]. Hence, the lubricating performance and the ability to escape chips reduce, leading to an increase in surface roughness. Moreover, $MoS_2$ tribofilm tends to form and grow when increasing the concentration of $MoS_2$ nanoparticles from 0 to 0.5%, thereby playing an important role in lubricating the cutting zone, helping to reduce the coefficient of friction and cutting forces. However, this tribofilm tends to be unstable and is lost when the concentration is further increased to 0.8%, as reported in the work of B. Rahmati et al. [25].

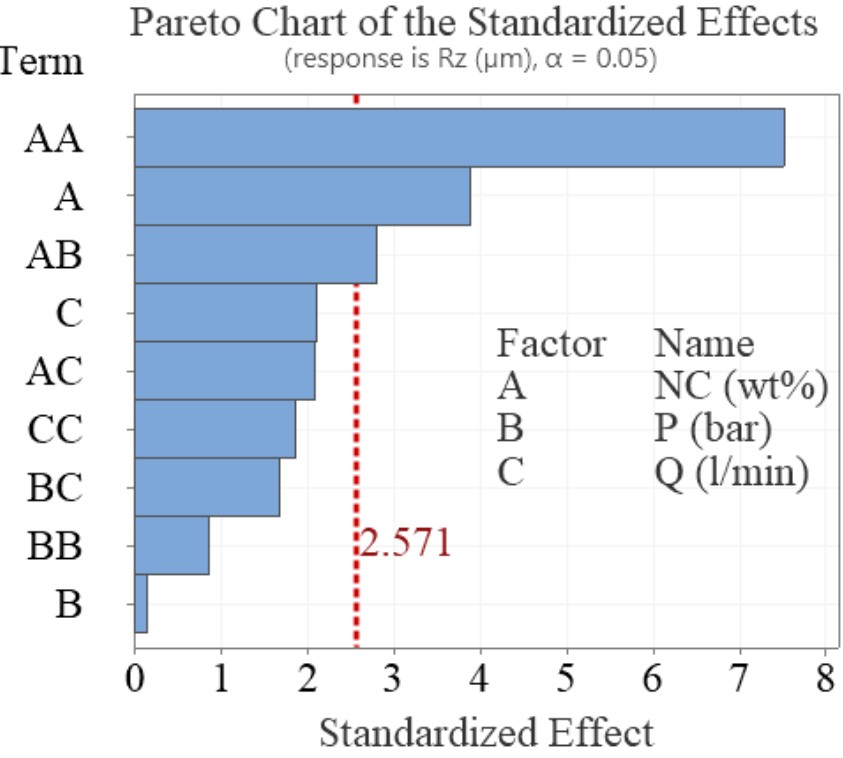

**Figure 3.** Pareto chart of effects of input machining factors on surface roughness, $R_z$. (A is *NC*: nanoparticle concentration, B is *P*: air pressure, C is *Q*: air flow rate.)

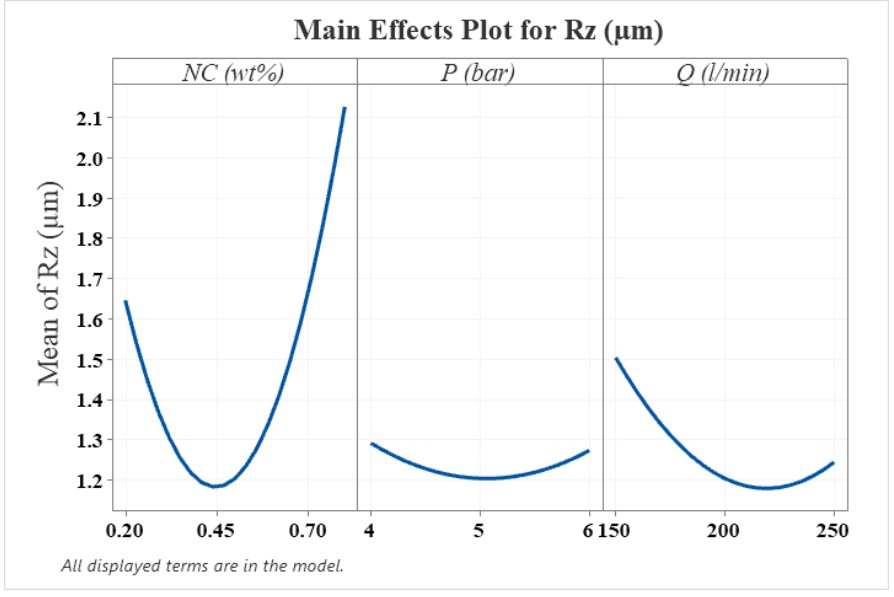

**Figure 4.** Main effects of input machining factors on the mean values of surface roughness, $R_z$ (*NC*: nanoparticle concentration, *P*: air pressure, *Q*: air flow rate).

From Figure 4, it can be also observed that the surface roughness values do not change much when increasing the air pressure from 4 bar to 6 bar, and this is the ideal pressure range used in production workshops. In addition, the surface roughness values also decreased gradually when reducing the air flow rate from 150 to 220 L/min and tended to grow slightly as Q increased to 250 L/min. The increase in the air flow will contribute to bring more cutting oil into the cutting area, thereby improving the lubrication performance [30,31].

The influence of nanoparticle concentration and air pressure on surface roughness in the case of a fixed air flow rate of 200 L/min is shown in Figure 5. It is recommended to choose a nanoparticle concentration around the value of 0.4 % and air pressure of less than 5 bar to achieve the smallest surface roughness values (<1.2 μm). The MQL method shows superior ability in delivering cutting oil in the form of high-pressure oil mist into the cutting zone and significantly improves the lubricating capacity compared to the flood condition [31]. However, the cutting space in turning is an open space, so the use of air pressure plays a very important role. If the air pressure is too high, the cutting oil will be pushed out of the cutting area and make it difficult to form oil mist on the contact face.

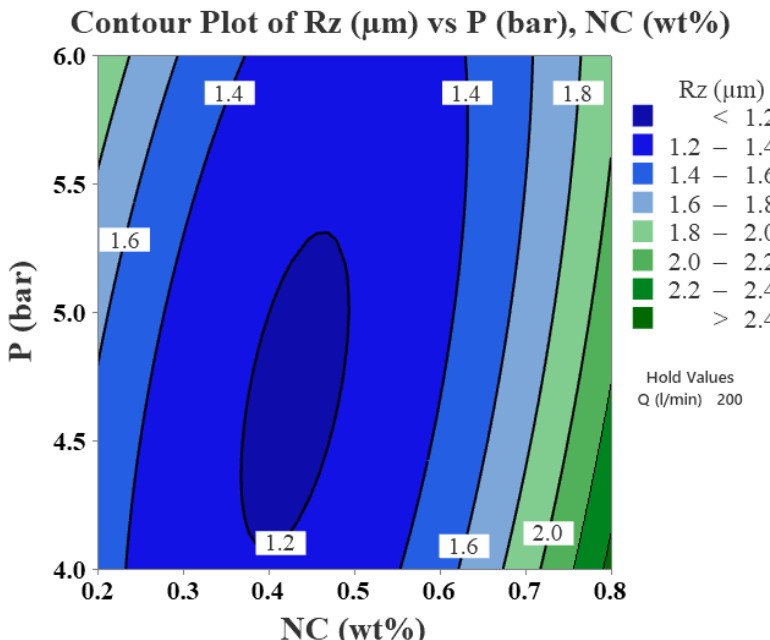

**Figure 5.** Contour plot of effects of nanoparticle concentration and air pressure on surface roughness with Q = 200 L/min.

### 3.2. Effects of NF MQL Parameters on Resultant Cutting Force

The resultant cutting force, $F_r$, is one of the important parameters of the cutting process, which directly affects the surface quality, tool life, and machining productivity. ANOVA analysis with 95% confidence was performed for the resultant cutting force, $F_r$, and the results are shown in Table 5. The degree of influence of the factors on the objective function of the resultant cutting force, $F_r$, and the fit level of the model is evaluated through the Fisher coefficient and P coefficient. The nanoparticle concentration (F = 33.67 and $p$ = 0.002) and the second-order interaction of the nanoparticle concentration (F = 67.09 and $p$ = 0) have a large F coefficient and a very small $p$-value compared with 0.05, thus greatly affecting the value of the objective function $F_r$. Other factors and interactions have little influence on the value of the resultant cutting force, $F_r$. The influence of the survey variables and their interactions on the resultant cutting force is also shown in the Pareto chart (Figure 6).

**Table 5.** Result of ANOVA analysis for resultant cutting force, $F_r$.

| Source | DF | Adj. SS | Adj. MS | F-Value | *p*-Value |
|---|---|---|---|---|---|
| Model | 9 | 516,799 | 57,422 | 11.69 | 0.007 |
| Linear | 3 | 166,178 | 55,393 | 11.27 | 0.012 |
| NC (wt%) | 1 | 165,441 | 165,441 | 33.67 | 0.002 |
| P (bar) | 1 | 701 | 701 | 0.14 | 0.721 |
| Q (L/min) | 1 | 36 | 36 | 0.01 | 0.935 |
| Square | 3 | 342,414 | 114,138 | 23.23 | 0.002 |
| NC (wt%)*NC (wt%) | 1 | 329,613 | 329,613 | 67.09 | 0.000 |
| P (bar)*P (bar) | 1 | 2047 | 2047 | 0.42 | 0.547 |
| Q (L/min)*Q (L/min) | 1 | 11,319 | 11,319 | 2.30 | 0.190 |
| 2-Way Interaction | 3 | 8207 | 2736 | 0.56 | 0.666 |
| NC (wt%)*P (bar) | 1 | 6698 | 6698 | 1.36 | 0.296 |
| NC (wt%)*Q (L/min) | 1 | 1055 | 1055 | 0.21 | 0.663 |
| P (bar)*Q (L/min) | 1 | 454 | 454 | 0.09 | 0.773 |
| Error | 5 | 24,566 | 4913 | | |
| Lack-of-Fit | 3 | 24,363 | 8121 | 80.04 | 0.012 |
| Pure Error | 2 | 203 | 101 | | |
| Total | 14 | 541,365 | | | |

**Figure 6.** Pareto chart of effects of input machining factors on resultant cutting force $F_r$ (A is *NC*: nanoparticle concentration, B is *P*: air pressure, C is *Q*: air flow rate).

The regression model for the resultant cutting force is given by Equation (3). The coefficient of determination $R^2 = 95.46\%$ and the adjusted coefficient of determination $R^2 = 87.29\%$ are quite large, which proves that the regression model for predicting the resultant cutting force is suitable with the experimental data. The results of the ANOVA analysis in Table 5 show that the Fisher coefficient (F-Value) calculated for the survey model is large (11.69) and the probability value $p = 0.007$ is less than 0.05, showing that the selected quadratic model is suitable, and there is only a 0.01% chance that this model is still affected by noise.

$$F_r \text{ (N)} = 315 - 1942{*}NC + 356{*}P - 7.21{*}Q + 3320{*}NC {*}NC - 23.5{*}P{*}P + 0.0221{*} Q{*}Q - 136{*}NC {*}P - 1.08{*}NC {*}Q - 0.213{*}P{*}Q \quad (3)$$

Figure 7 shows a graph of the influence of the survey variables on the resultant cutting force, $F_r$. The graph of the influence of the nanoparticle concentration on $F_r$ has an inflection point, showing that the investigated range is appropriate.

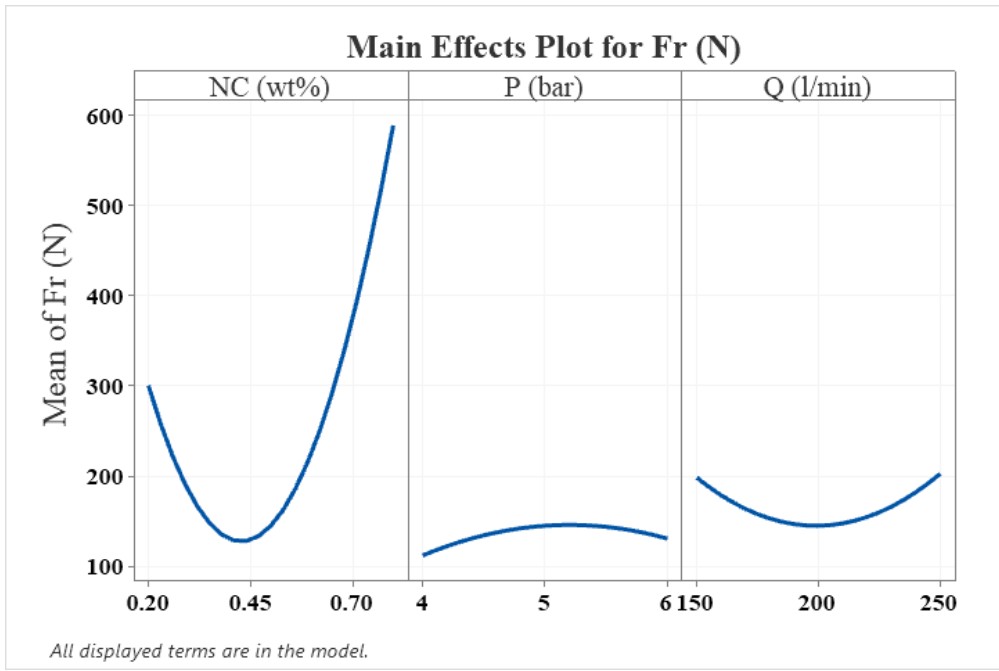

**Figure 7.** Main effects of input machining factors on the mean values of the resultant cutting force, $F_r$.

From Figure 8, the resultant cutting force decreases gradually with increasing the nanoparticle concentration from 0.2% to 0.45%, but it increases when increasing the NC to 0.8%. The reason is that the turning process is a machining process with an open cutting area and a large dynamic rear angle, so a part of the nano-cutting oil is dissipated into the environment, and the other will escape along the rake face, facilitating the chip to escape easily. At the same time, $MoS_2$ nanoparticles possess a good lubricating ability and have a sheet structure with a large surface area, so with a reasonable concentration the nanoparticles easily adhere to the surface creating favorable conditions for the formation of tribofilm, thereby reducing friction and cutting forces. In addition, $MoS_2$ nanosheets have high thermal conductivity, so when mixed with vegetable oil, they improve the cooling capability of the base cutting oil [31]. However, when the nanoparticle concentration is increased, the nanoparticles clump and interfere with the chip escape and heat dissipation, thus increasing the cutting force due to the easily adhesive properties of $MoS_2$ nanosheets [12,25].

It can be also observed that the cutting force changed very little with the increase in the air pressure from 1 to 6 bar. However, the air flow rate also affects the resultant cutting force. The resultant cutting force reached the minimum when using an air flow rate of about 200 L/min.

The influence of the nanoparticle concentration and air pressure on the resultant cutting force, $F_r$, with Q fixed at 200 L/min, was investigated and is shown in Figure 8. Based on the obtained results, the reasonable values of NC and P can be rapidly chosen for the expectation of a smaller $F_r$. For Q= 200 L/min, it is recommended to choose a nanoparticle concentration around 0.4% and air pressure of less than 4.5 bar to achieve a resultant cutting force of less than 100 N.

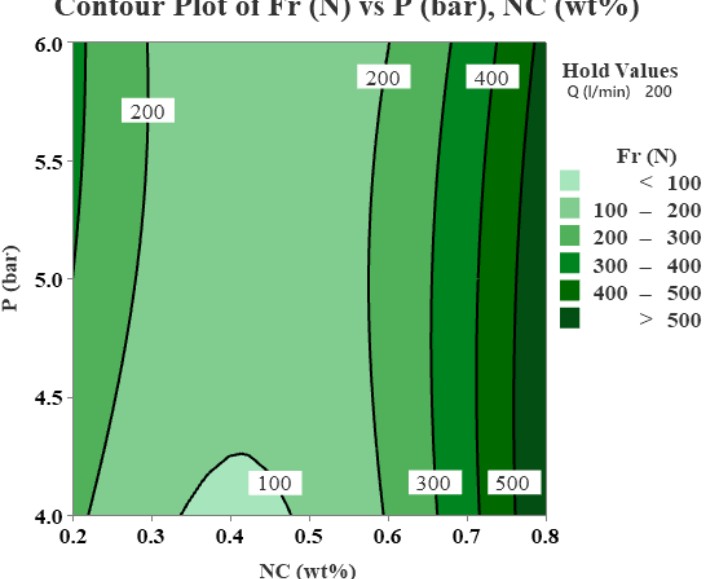

**Figure 8.** Contour plot of effects of nanoparticle concentration and air pressure on resultant cutting force with Q = 200 L/min (cutting speed, V = 160 m/min; depth of cut, t = 0.12 mm; and feed rate, f = 0.12 mm/rev).

### 3.3. Determination of Optimal Air Pressure, Air Flow Rate, and Nanoparticle Concentration

Based on the specific conditions and requirements of the machining process, it is possible to choose the appropriate objective function and optimal criteria. For a better machined surface quality (in finishing), single-objective optimization should be used and the results are shown in Figure 9. The surface roughness reaches the minimum value, $R_z$ = 1.1354 μm, with NC = 0.42%, P = 4.14 bar, and Q = 230 L/min.

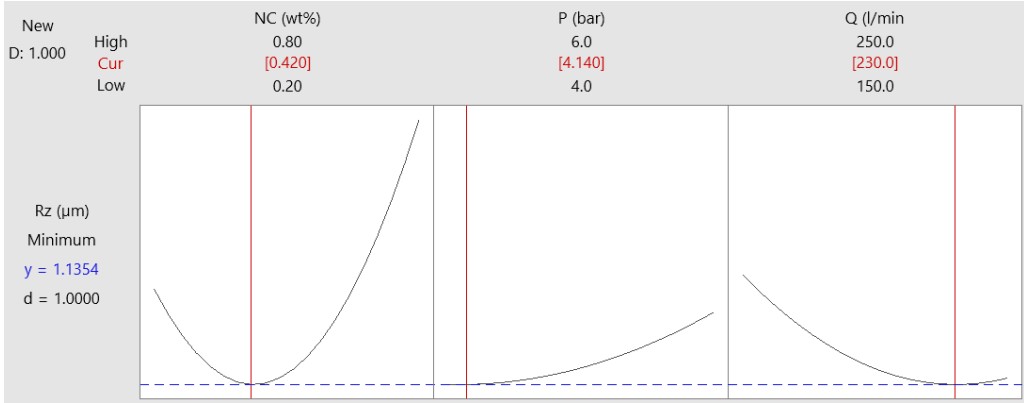

**Figure 9.** Single-objective optimization for $R_z$ (cutting speed, V = 160 m/min; depth of cut, t = 0.12 mm; and feed rate, f = 0.12 mm/rev).

For the minimal resultant cutting force, single-objective optimization of $F_r$ should be used and the results are shown in Figure 10. The predicted lowest cutting force, $F_r$, is 82.17 N at NC = 0.4%, P = 4.0 bar, and Q = 192 L/min.

For the goal of better surface quality and smaller cutting force, the multi-objective optimization of $R_z$ and $F_r$ was carried out and the obtained results are shown in Figure 11. By using NC = 0.42%, P = 4.14 bar, and Q = 211 L/min, the lowest predicted values were $F_r$ = 113.67 N and $R_z$ = 1.1587 μm.

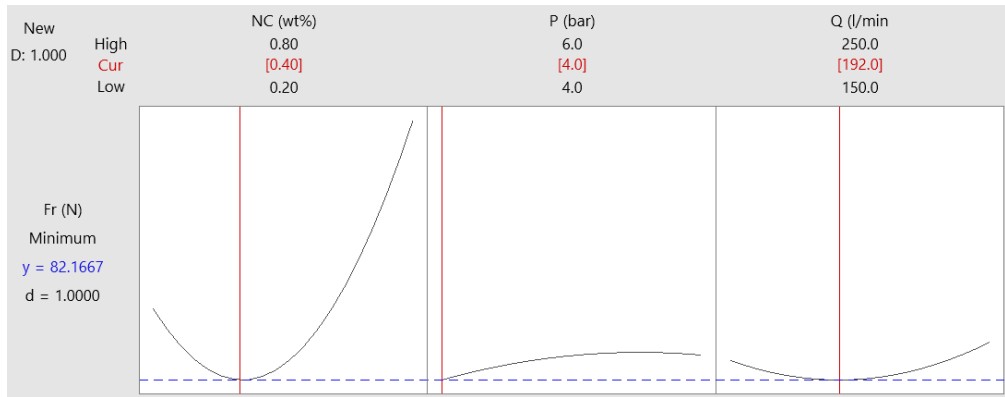

**Figure 10.** Single-objective optimization for $F_r$ (cutting speed, V = 160 m/min; depth of cut, t = 0.12 mm; and feed rate, f = 0.12 mm/rev).

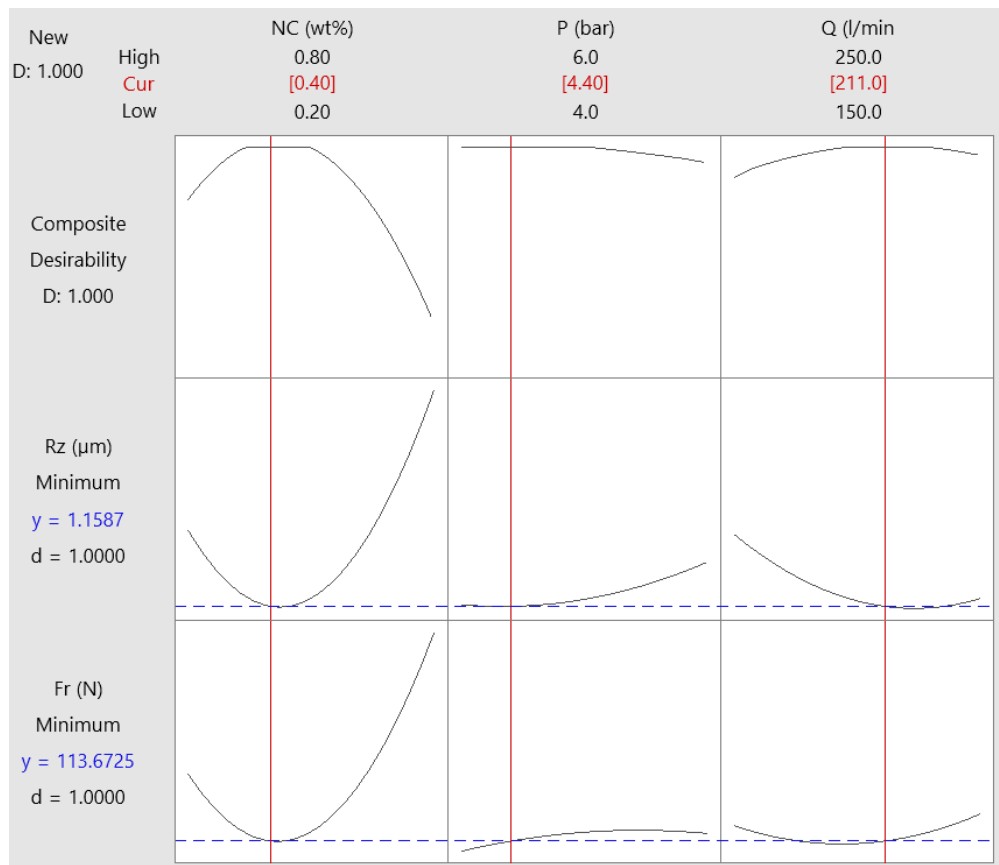

**Figure 11.** Multi-objective optimization for $R_z$ and $F_r$ (cutting speed, V = 160 m/min; depth of cut, t = 0.12 mm; and feed rate, f = 0.12 mm/rev).

## 4. Conclusions

This work has successfully applied MQL using $MoS_2$ nano-cutting oil. Using $MoS_2$ nanoparticles suspended in rice bran oil, an environmentally friendly vegetable oil, in the hard turning process has great significance, not only in terms of technology but also from an environmental point of view. The enhancement in lubrication and cooling in the cutting zone has improved the cutting efficiency, machined surface quality, and tool machinability. A Box-Behnken design of experiment and ANOVA analysis were applied to analyze the influence of the technological parameters including $MoS_2$ nanoparticle concentration, air flow pressure, and air flow rate on the surface roughness, $R_z$, and the resultant cutting force, $F_r$. Some findings can be summarized as below:

- The nanoparticle concentration and the interaction effect between NC and P have the greatest influences on the surface roughness. Meanwhile, the air pressure has a great impact on the resultant cutting force.
- From the analysis of the contour plots, it is possible to select a reasonable range of values for the investigated parameters to achieve the smallest $R_z$ or $F_z$. Values of $Q = 200$ L/min, NC of about 0.4%, and air pressure of less than 4.5 bar should be used to achieve the smallest $R_z$ (<1.2 μm) and $F_r$ (<100 N).
- For achieving the smallest surface roughness, $R_z$, the predicted minimum $R_z$ is 1.1354 μm with NC = 0.42%, P = 4.14 bar, and Q = 230 L/min. For achieving the smallest cutting force, $F_r$, the predicted lowest $F_r$ is 82.17 N at NC = 0.4%, P = 4.0 bar, and Q = 192 L/min.
- Through implementing multi-objective optimization, the optimal set NC = 0.42%, P = 4.14 bar, and Q = 211 L/min should be used to obtain the smallest $R_z$ or $F_z$ values.

In further studies, more investigation is needed on the microstructure of the machined surface and the lubricating mechanism of $MoS_2$ nano-cutting oil with the proposed optimal technological factors.

**Author Contributions:** Conceptualization, N.M.T. and T.T.L.; Methodology, N.M.T., T.T.L. and T.B.N.; Software, T.T.L. and T.B.N.; Validation, N.M.T. and T.B.N.; Formal analysis, N.M.T., T.T.L. and T.B.N.; Investigation, T.T.L. and T.B.N.; Data curation, T.B.N.; Writing – original draft, N.M.T., T.T.L. and T.B.N.; Writing – review & editing, N.M.T. and T.T.L.; Supervision, N.M.T., T.T.L. and T.B.N.; Project administration, N.M.T. All authors have read and agreed to the published version of the manuscript.

**Funding:** This research was funded by Thai Nguyen University of Technology, Thai Nguyen University, Vietnam.

**Acknowledgments:** The work presented in this paper is supported by Thai Nguyen University of Technology, Thai Nguyen University, Vietnam.

**Conflicts of Interest:** The authors declare no conflict of interest.

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
