# Peer review of "Study of Effects of MoS2 Nanofluid MQL Parameters on Cutting Forces and Surface Roughness in Hard Turning Using CBN Insert"

_fluids, doi:10.3390/fluids8070188_

Round 1

Reviewer 1 Report

I believe the article titled Study of effects of MoS2 nanofluid MQL parameters on cutting forces and surface roughness in hard turning using CBN insert has presented appreciable work. However for enhancing the quality of the article following comments should be addressed in the article.

1. In the introduction a paragraph must be added to discuss the other optimization techniques too. Like ANN, GA, etc. How is the present optimization technique beneficial over the others?

2. Why the Box Behken Design of the experiment was used in the current research? Any particular reason for that?

3. A flow chart showing the full methodology of the presented technique will be useful for the readers.

4. A table that includes all the details (minor and major) about the experimentation and parameters should be added to the paper.

5. The Conclusion should be precise and should be aligned with the objectives of the paper.

6. The Discussion should be more elaborated.

7. Why the thermophysical properties were not investigated of the nano fluid?

8. The references are too short for this article. There are many papers that should have been cited but somehow overlooked by the authors. For eg.

https://www.sciencedirect.com/science/article/abs/pii/S0301679X23001160

It should be improved. 

Author Response

RESPONSES TO THE REVIEWER 1

We are very grateful for the reviews provided by the editors and each of the external reviewers of this manuscript. Please see below, our detailed response to comments.

  1. In the introduction a paragraph must be added to discuss the other optimization techniques too. Like ANN, GA, etc. How is the present optimization technique beneficial over the others?

Answer:

Thank you very much for your very useful comments. The other optimization techniques like ANN, GA, etc. were added and discussed in the introduction. Also, the characteristics and application of these techniques were analyzed. The advantages of the method used in the paper were added in the revised manuscript.

  1. Why the Box Behken Design of the experiment was used in the current research? Any particular reason for that?

Answer:

Thank you very much for your very useful comments. Currently, there are many methods used to analyze and optimize machining processes. Many studies have used the optimization algorithms like ANN, GA, and so on. to provide the prediction for regression function and determine the optimal set of parameters. However, these methods do not evaluate the influence of the survey variables and their interaction effects on the objective function. Besides, optimization methods using experimental planning models such as Taguchi, Box-behnken, Factory, CCD, etc. are commonly used in experimental research because they can reduce experiments and still evaluate the influence of the factors on the objective function. At the same time, these methods can also be used to find predictive models and determine the optimal set of parameters for the objective function. The discussion was added in the revised manuscript.

  1. A flow chart showing the full methodology of the presented technique will be useful for the readers.

Answer:

Thank you very much. The authors analyzed and discussed the methodology commonly used in studying the influence of parameters on the cutting performance, thereby showing the suitability and reasons for using the Box-Behnken optimal planning method in this study. Please allow the author to be cited in the revised manuscript. Thank you again

  1. A table that includes all the details (minor and major) about the experimentation and parameters should be added to the paper.

Answer:

Thank you very much. The information about the experimental parameters was added to the table 3 by following the reviewer’s comment.

  1. The Conclusion should be precise and should be aligned with the objectives of the paper.

Answer:

Thank you very much. The Conclusion was rewritten to precisely show the main results.

  1. The Discussion should be more elaborated.

Answer:

Thank you very much. The Discussion was expanded

  1. Why the thermophysical properties were not investigated of the nano fluid?

Answer:

Thank you very much. The measurement of thermal conductivity of nanoparticles suspended in vegetable oil has been conducted by many studies. Please allow the authors to use the citation (Please see Ref. 31) and was added in the revised manuscript.

  1. The references are too short for this article. There are many papers that should have been cited but somehow overlooked by the authors. For eg.

https://www.sciencedirect.com/science/article/abs/pii/S0301679X23001160

Answer:

Thank you very much. The references were expanded and the suggested article was revised carefully and cited in the revised manuscript.

Reviewer 2 Report

My comments are in the file attached.

Author Response

RESPONSES TO THE REVIEWER 2

We are very grateful for the reviews provided by the editors and each of the external reviewers of this manuscript. Please see below, our detailed response to comments.

Lubrication and cooling play a vital role in hard machining and have become increasingly important. The
selection of an appropriate cooling lubrication method significantly impacts machining efficiency and
the quality of the machined surface. One innovative solution that has shown promising results in hard
turning is the utilization of minimum quantity lubrication (MQL) with nano-cutting oils. This study
focuses on examining the effects of various MQL technological parameters, such as nanoparticle
concentration, air pressure, and air flow rate, on surface roughness and cutting force during hard
turning operations employing CBN inserts. The aim is to investigate the efficacy of MQL with MoS2
nano-cutting oil in enhancing these machining aspects. Ideas were Ok, but formal revision of literature
was really poor.
State of the art: por really, many fundamental works missed, as it was the original ideas of
https://doi.org/10.1016/j.jclepro.2017.07.078 and other from interesting researchers like Lopez de
Laclle or A Rodriguez who merge the knowledge with experimental results in superalloys and titanium.
More than 10 works, missed all. Some of your results must be discussed in comparison with missed
works.

Answer:

Thank you very much. The state of the art from the very important work of leading researchers like Lopez de

Laclle or A Rodriguez was carefully studied, revised and cited (Refs. 3,6,29,30) in the introduction section in the revised manuscript.

Harmful particles in the nanofield can be dangerous for human being. Did you check this aspects. This is
under the real stopper in many industries for using nanoparticles.
Answer:

Thank you very much for your valuable comment. In this issue, the authors also checked the research results on the level of impact on human health. The MoS2 nanoparticles used in this paper belong to the group of nanoparticles that have the least impact on human health (https://doi.org/10.1016/j.geoen.2023.211767). Actually in the experiment, the authors also found that it is necessary to use the ventilation fans and exhaust fans for cutting oil containing nanoparticles in the mist form to minimize the harm to users. This issue should be further considered.

Anova is OK, but please reduce its explanation.
Figure 1 is not well explained, in 3D world the MQL goes in many directions. This was explained by CFD
in Experimental and numerical investigation of the effect of spray cutting fluids in high speed milling,
Journal of Materials Processing Technology 172 (1), 11-15
Answer:

Thank you very much. The work was revised carefully and cited in the revised manuscript in order to improve the paper quality.

Table 1 can be eliminated. Are they standard values?
Answer:

Thank you very much. Table 1 is only a table showing the percentages of the elements in the steel used in the experiments. The author gives it back so that the readers can easily follow or look up the equivalent material. If necessary, the author can remove this table.

Eq1…too simple….Lamikiz, or Urbikain, or Olvera disused that the real interesting subcomponent could
be only one of combination of 2 of them. Tangential force (along Vc) is the main one.

Answer:

Thank you very much for your valuable comment. In the study of cutting forces, there are 3 components, Fx, Fy, Fz, in which the thrust force Fy and tangential force Fz are the main forces and have their own characteristics, and the authors has conducted the study to investigate these force components. However, the purpose of the paper is to study and evaluate the influence of technological parameters including nanoparticle concentration, air pressure and air flow rate, so here, the survey is a general orientation. Thereby, the author surveyed Fr, thereby determining the influence trend and reasonable value domain of the input variable served as the basis for further deeper studies on the influence on the cutting force components.

Roughness is OK, again ANOVA must be reduced, many charts are meaningless. Give conclusions as
points: each idea one point.

Answer:

Thank you very much. Some charts were removed and ANOVA was reduced. The Conclusion was rewritten to precisely show the main results in the revised manuscript.

Check works by leading groups, some above but for instance see
https://doi.org/10.1177/0954406215616145 because the main resercah work in experimental field do
not use Taguchi or Anova. Machining is a physical process, use the basis and fundamentals

Answer:

Thank you very much for extremely valued work. The authors were revised carefully and improve the paper quality from the basis and fundamentals for hard machining in the Introduction in the revised manuscript.

Round 2

Reviewer 1 Report

Can be accepted. 

Moderate editing is required 

Reviewer 2 Report

My suggestions were taken into account. Accepted.